# Isotope-Based Techniques to Investigate Factors Influencing Water Use Efficiency in *Pinus koraiensis* Leaves during Plant Growth

**DOI:** 10.3390/plants13131771

**Published:** 2024-06-27

**Authors:** Tiantian Fang, Guangze Jin, Zhili Liu

**Affiliations:** 1Center for Ecological Research, Northeast Forestry University, Harbin 150040, China; 17750271724@163.com (T.F.); taxus@126.com (G.J.); 2Key Laboratory of Sustainable Forest Ecosystem Management-Ministry of Education, Northeast Forestry University, Harbin 150040, China; 3Northeast Asia Biodiversity Research Center, Northeast Forestry University, Harbin 150040, China

**Keywords:** δ^13^C, δ^18^O, plant water use efficiency, leaf functional traits, plant size

## Abstract

Plant water use efficiency (WUE) is a comprehensive physiological indicator of plant growth and ability to adapt to drought. However, research on the mechanisms controlling WUE during plant growth and development remains weak. Here, we studied *Pinus koraiensis* as a typical evergreen conifer species in Northeast China. After collecting 80 tree samples with varying diameters at breast height (DBH), we measured δ^13^C and δ^18^O as an indicator of WUE, leaf morphology (volume, dry weight, and total epidermal area), ecological stoichiometry (carbon, nitrogen, and phosphorus content), and abiotic factors (light environment, soil pH, soil water content, and soil nutrient content). Correlational analysis of these variables revealed distinct differences between smaller/younger and larger/older plants: (1) In plants with DBH less than 52 cm, δ^13^C was positively related to DBH, and δ^18^O was negatively related to DBH. Plants with DBH greater than 52 cm showed no relationship between δ^13^C and DBH, and δ^18^O was positively related to DBH. (2) In plants with DBH less than 52 cm, there was a negative correlation between δ^13^C and δ^18^O and between δ^13^C and leaf phosphorus content (LP), but a positive correlation between δ^13^C and DBH, leaf mass per area (LMA), and leaf density (LD). The slopes of DBH-δ^13^C, δ^18^O-δ^13^C, leaf nitrogen content (LN)-δ^13^C, and LMA-δ^13^C correlations were greater in smaller plants than large plants. (3) Structural equation modelling showed that in smaller plants, DBH had a direct positive effect on δ^13^C content and a direct negative effect on δ^18^O, and there was a direct positive effect of light environment on δ^18^O. In larger plants, there was a direct negative effect of light environment on δ^13^C and a direct positive effect of DBH on light environment, as well as a negative effect of soil nitrogen content on leaf nitrogen. In smaller plants, DBH was the most important factor influencing δ^13^C, followed by δ^18^O and soil moisture, with light and soil pH showing minimal influence. In larger plants, light environment influenced δ^13^C the most, followed by soil nitrogen content and soil moisture content, with leaf nitrogen and DBH contributing little. The results suggest that water use efficiency strategies of *P. koraiensis* vary according to growth stage, and the effects of abiotic factors and functional traits vary at different growth stages.

## 1. Introduction

Water is critical for plant life and plays an important role in plant growth, development, and distribution. A plant’s water use efficiency (WUE) score represents the ratio of carbon fixation per unit mass to water consumption through transpiration [1]. Since water is needed to form organic matter, WUE is an important physiological indicator of plant growth and ability to adapt to drought [2]. The WUE score illustrates the tradeoff between photosynthetic carbon fixation and water loss through transpiration and is a key physiological parameter linking carbon and water cycles [3].

The leaf is the primary site for energy conversion in plants, where light is intercepted and carbon is produced through photosynthesis [4,5]. Leaf traits vary according to resources such as light and water and therefore link plants to their environment [6,7]. Leaf economic spectrum is an interrelated and synergistic combination of leaf functional traits, and it also quantitatively represents a series of regular and continuously changing plant resource tradeoff strategies [8,9]. Importantly, leaf photosynthetic and transpiration traits regulate and control water use efficiency (WUE) [10,11].

Plants inevitably lose water through stomatal transpiration while photosynthetically fixing carbon [12]. The carbon isotope δ^13^C can be used to measure WUE in leaves [13,14]; however, leaf δ^13^C alone is not a good reflection of long-term environmental conditions. Since leaf δ^13^C is driven by the *C_i_*/*C_a_* ratio (intracellular versus atmospheric CO_2_), the presence of δ^18^O must be considered [15]. Prieto et al. [16] took 15 tree species in the Mediterranean as research objects to evaluate the relationship between δ^13^C and δ^18^O and leaf photosynthetic carbon sequestration and found that leaf carbon and oxygen isotopes were coupled to leaf morphology and nutrients, suggesting that δ^13^C and δ^18^O could be used as part of the leaf economic spectrum. In cases where leaf δ^13^C is primarily affected by transpiration and water consumption, the δ^13^C and δ^18^O content of leaves are positively correlated. By contrast, when leaf δ^13^C and δ^18^O are unrelated or negatively correlated, this indicates that leaf δ^13^C is primarily affected by photosynthetic carbon sequestration [1]. Thus, the combination of δ^13^C and δ^18^O can effectively describe the effects of photosynthetic carbon fixation, transpiration, and water consumption on WUE [15].

Currently, most studies on plant water use efficiency focus on environmental impact. Through meta-analysis, Mathias et al. (2021) found that the intrinsic WUE of trees is enhanced by a global increase in atmospheric CO_2_ levels, and concluded that WUE is regulated by climate as well as plant type [17]. Wang et al. (2021) used structural equation models to study the effects of soil properties, biodiversity, and leaf traits on δ^13^C and δ^18^O [18,19,20]. Some researchers have explored the relationship between plant WUE and the internal organizational structure of plant leaves at the anatomical level. For example, Trueba et al. (2021) found that leaf structure was the determining factor of WUE in conifers [21]. It is also widely reported that plant size is one of the main factors affecting intraspecific plant trait variation [22,23]. Changes in the ability of plants to absorb nutrients and adapt to environmental conditions during ontogenetic development may lead to changes in the structure and function of trees [24], thus affecting the WUE of plants. However, there are insufficient studies on the relationship between WUE and plant size and the underlying causes.

*Pinus koraiensis* is a constructive species in the broad-leaved and Korean pine forests of the mountainous area in Northeast China, with high economic and ecological value [25,26]. However, due to human interference, such as excessive cutting and beating for pine nuts, the number of existing plants has declined sharply. Therefore, it is of great significance to study the mechanisms controlling WUE in *P. koraiensis* during its growth in order to maintain the stability of the forest ecosystem. In this study, 80 *P. koraiensis* trees of different sizes were randomly selected from a typical *P. koraiensis* forest in the Lesser Khingan Mountains, and the quantity of δ^13^C and δ^18^O in the needles of *P. koraiensis* was determined. To evaluate factors influencing WUE among different plant sizes, we measured leaf morphological traits such as leaf volume, dry weight, and total epidermis area, as well as ecological stoichiometric characteristics such as carbon, nitrogen, and phosphorus content, and abiotic factors including light environment, soil pH, soil moisture, and soil nutrient content. The results provide a scientific understanding of water use regulation during the growth of *P. koraiensis*.

## 2. Results

### 2.1. Trends in Leaf δ^13^C and δ^18^O as a Function of Plant Size

Overall, δ^13^C content fluctuated in the range of −32.192‰ to −27.533‰, with a mean value of −29.223‰ (Figure 1A). Values for δ^18^O fluctuated between 18.403‰ and 23.470‰, with a mean value of 20.283‰ (Figure 1B). At a DBH of 52 cm, δ^13^C and δ^18^O show an inflection point (Figure 1). When the DBH was less than 52 cm, there was a positive relationship between δ^13^C and DBH, whereas a negative relationship was observed for δ^18^O. When the DBH was greater than 52 cm, δ^13^C did not change with DBH, whereas δ^18^O was positively related to DBH (Figure 1B).

### 2.2. Relationship between δ^13^C in P. koraiensis Leaves and Related Leaf Functional Traits and Abiotic Factors

Based on the relationships identified between δ^13^C and δ^18^O and DBH, the data set was divided into two groups (DBH above or below 52 cm) and correlations between carbon and oxygen isotopes and plant traits were determined. For samples with DBH less than or equal to 52 cm (Table 1), δ^13^C and δ^18^O were negatively correlated with one another (*p* < 0.01). Correlational analysis between δ^13^C and various plant traits revealed a negative relationship between δ^13^C and leaf phosphorus content (LP) (*p* < 0.05) but a positive relationship between δ^13^C and DBH, (*p* < 0.01), leaf mass per area (LMA) (*p* < 0.01), and leaf density (LD) (*p* < 0.05). For δ^18^O, there was a negative correlation with DBH and LMA (*p* < 0.01) and a positive correlation with LP (*p* < 0.01). There was no relationship between δ^13^C or δ^18^O and any of the abiotic factors analyzed (Table 2). 

For samples with DBH above 52 cm, δ^13^C and δ^18^O were not correlated with one another (Table 3). While δ^13^C was not correlated to any leaf traits, δ^18^O was negatively correlated with DBH (*p* < 0.01) and positively correlated with LC and LP (*p* < 0.05). In terms of abiotic factors (Table 4), δ^18^O was positively correlated with soil pH (*p* < 0.01), and negatively correlated with soil moisture content and soil carbon content (*p* < 0.01).

The slope of the correlation between plant traits and leaf δ^13^C was significantly different (*p* < 0.01) between the two size groups (Figure 2). The correlation slopes of DBH-δ^13^C, δ^18^O-δ^13^C, LN-δ^13^C, and LMA-δ^13^C were higher in plants with DBH less than or equal to 52 cm than in plants with DBH greater than 52 cm.

### 2.3. Influencing Factors of δ^13^C in P. koraiensis Leaves

The results of structural equation modelling showed that the best factor predicting δ^13^C was different between the two groups. For trees with DBH below 52 cm, the best predictors of leaf δ^13^C were δ^18^O and soil pH, while the best predictors of leaf δ^13^C for trees with DBH above 52 cm were leaf nitrogen content and soil nitrogen content (Figure 3).

A generalized linear mixed-effects model was used to determine the relative importance of each variable on δ^13^C variation in *P. koraiensis* leaves. For trees with DBH below 52 cm (Figure 3A), there was a direct positive effect of DBH on δ^13^C and a direct negative effect of DBH on δ^18^O (*p* < 0.01). In this group, DBH contributed the most to δ^13^C, followed by δ^18^O and soil moisture, while light and soil pH had little influence on δ^13^C (Figure 4A). For plants with DBH above 52 cm (Figure 3B), there was a highly negative effect of light environment on δ^13^C (*p* < 0.01), a direct positive effect of DBH on light environment (*p* < 0.01), and a significantly negative effect of soil nitrogen content on leaf nitrogen (*p* < 0.01). In this group, light environment contributed the most to δ^13^C, followed by soil N content and soil water content, while leaf N and DBH contributed minimally to δ^13^C (Figure 4B).

## 3. Discussion

### 3.1. Variation in the Water Use Efficiency of P. koraiensis According to Plant Size

For *P. koraiensis* plants with a DBH under 52 cm, leaf δ^13^C content was higher with increasing DBH, indicating greater water use efficiency (WUE). Plants with a DBH greater than 52 cm, which are considered mature [27,28,29], showed no obvious changes in WUE with increasing DBH. This suggests that WUE increases during the growth and development of *P. koraiensis* then stabilizes upon maturation. This may be explained by height-dependent changes in WUE that arise during plant growth, which also influence leaf functional traits [30,31]. According to the hydraulic limitation hypothesis of tree growth, the height of a plant changes minimally once mature. The δ^13^C content of *P. koraiensis* needles in our study (−32.19‰ to −27.53‰) fell within the global dataset (−37.83‰ to −19.85‰) [20]. However, the δ^13^C of *P*. needles (−29.86 ± 2.32‰) was poorer than that of the global dataset (−28.68 ± 2.68‰) [20] on the whole, which might suggest that *P. koraiensis* favors a water-consuming strategy. Since leaf δ^13^C was gradually enriched during the entire growth process of *P. koraiensis*, theoretically, the water utilization strategy of *P. koraiensis* should change from a water consumption strategy to water-saving strategy with increasing DBH.

In plants with DBH under 52 cm, leaf δ^18^O was gradually reduced with increasing DBH, while the opposite was true for plants with above 52 cm DBH. Since leaf δ^18^O can provide integrated information on the long-term stomatal behavior of leaves [1,32], these trends suggest that the transpiration effect of *P. koraiensis* varies with growth, i.e., transpiration water consumption decreases during the growth period and increases during the maturity period.

### 3.2. Strategies for Water Use Efficiency at Different Stages of P. koraiensis Growth

Others have shown that a positive correlation between leaf δ^13^C and δ^18^O indicates that leaf δ^13^C content is primarily affected by transpiration and water consumption [1]. By contrast, when leaf δ^13^C and δ^18^O are not correlated or are negatively correlated, this suggested that leaf δ^13^C is primarily affected by photosynthetic carbon sequestration [1]. Our experimental results showed a negative correlation between δ^13^C and δ^18^O in the lower DBH group and no correlation in the higher DBH group, suggesting that the WUE of *P. koraiensis* is primarily affected by photosynthetic carbon sequestration across all growth and development stages.

Plants achieve functional diversity through the combinatorial relationships of multiple trait assemblages [12]. Since both photosynthetic carbon fixation and transpiration water consumption are affected by leaf traits, the relationship between leaf photosynthetic carbon fixation and transpiration water consumption traits assemblages can influence long-term WUE [10,11]. Leaf δ^13^C and leaf mass per area (LMA) were positively correlated in younger plants with low DBH, suggesting that LMA plays a positive role in regulating plant WUE. Others have found that LMA is the main influencing factor of δ^13^C variation and that leaf structure (thickness, tissue density, and LMA) affects the ability of mesophyll conductance to control photosynthetic CO_2_ supply [33,34]. At the same time, LMA affects the distribution of N and P in leaves and has a regulatory effect on the photosynthetic capacity of plants [7,35].

The relationship between leaf δ^13^C and phosphorus may be positively correlated [36], negatively correlated [37], or unrelated [38]. The present study showed that P content in leaves is negatively correlated with δ^13^C at the low DBH stage but not at the high DBH stage, which indicates that the uptake of soil P by plant roots in the growth and development stage may be related to transpiration, while the long-term WUE of plants at the mature stage may be regulated by water consumption for transpiration rather than photosynthetic carbon sequestration. This seems to contradict the previous statement; however, we showed that leaf phosphorus content (LP) has a negative relationship to DBH in the growth and development stage (DBH < 52 cm) and has no correlation with DBH in the maturity stage (DBH > 52 cm). In general, shorter trees tend to allocate more P to the leaves to improve carbon capture efficiency, while taller trees may need to allocate more P to the roots [39]. Therefore, during the growth and development of *P. koraiensis*, the distribution of phosphorus may gradually shift from the leaves to the roots, and when the trees mature, the P content in the leaves may stabilize, just like for δ^13^C. The correlation between LP and δ^13^C was not significant, which therefore cannot indicate that long-term WUE is regulated by transpiration water consumption in the mature stage of *P. koraiensis*.

### 3.3. Factors Influencing δ^13^C in P. koraiensis Leaves

Our results showed that the WUE of *P. koraiensis* is regulated by multiple factors and varies according to tree size. For trees with DBH less than 52 cm, DBH had a direct positive effect on leaf δ^13^C but had no direct effect on trees with DBH greater than 52 cm. For trees with DBH under 52 cm, light had no direct effect on δ^13^C but had a negative effect on trees with DBH above 52 cm. In addition, with tree growth, the most influential leaf traits predicting leaf δ^13^C shifted from δ^18^O to leaf N, and the most influential soil chemical properties shifted from soil pH to soil N, indicating that the regulatory mechanisms controlling WUE varies with the size of *P. koraiensis*.

In the smaller DBH group of *P. koraiensis*, DBH contributed the most to δ^13^C leaf content, followed by δ^18^O and soil moisture. This indicates that plant size is an important driving force for changes in WUE during the growth and development of *P. koraiensis*. As the plant grows, they adopt water-saving strategies such as reducing transpiration water consumption to maintain a high WUE. Unsurprisingly, light had a significant effect on δ^18^O, because light is directly related to transpiration and water consumption characteristics such as leaf transpiration rate and stomatal conductance [40,41].

In older plants with DBH greater than 52 cm, DBH had a direct positive effect on light environment, while light environment had a direct negative effect on δ^13^C. Light environment usually only contributes positively to the growth of young trees [42,43,44]. When photosynthetically active radiation reaches a certain intensity, the light saturation point of *P. koraiensis* is reached. At this point, photosynthesis cannot increase with the increase in light environment [45,46], and WUE decreases with increases in photosynthetically active radiation. Due to the strong light received by the upper leaves of large trees, the frequency of stomatal closure increases with strong transpiration, and water stress increases [47]. There exists, therefore, a negative relationship between WUE and light environment. We also showed that soil N content had a negative effect on leaf N, which may be because at the mature stage, soil N content was no longer a limiting factor for plant growth [48] and plant N use efficiency decreased. At the same time, due to the opposite stoichiometric characteristics of large trees above the forest and plants below the forest, soil nutrients are preferentially absorbed by plants under the forest [49]; so, this result may be caused by the fact that most of the N decomposed by coniferous litter in the soil is absorbed by herbs and shrubs.

Our data revealed that DBH-δ^13^C, δ^18^O-δ^13^C, LN-δ^13^C, and LMA-δ^13^C correlations showed allometric growth relationships, which confirms that plant size may affect some models of leaf trait–trait relationships [50]. Compared with the mature stage of plant growth and development, the relationship between WUE and photosynthetic traits (LN and LMA) and transpiration traits (δ^18^O) was more coupled in growth plants.

## 4. Materials and Methods

### 4.1. Site Profile

The field investigation was conducted in the Liangshui National Nature Reserve of Heilongjiang Province (47°10′50″ N, 128°53′20″ E), which is located on the southern slope of the Lesser Khingan Mountains. The climax vegetation is the broad-leaved *P. koraiensis*, and the total forest stock is (1.88 × 10^6^) m^3^ with an altitude of 280–707 m. The average slope is between 10 and 15°, representing a low-mountain and hilly landform. The climate type is continental monsoon, with little precipitation and high wind in spring. Summer is short and humid with high temperatures; in autumn, the temperature drops sharply, and early frost often occurs. Winter is long and dry, cold, and snowy. The average annual temperature is −0.3 °C, the average annual precipitation is 676 mm, and the average annual evaporation is 805 mm. The average annual precipitation days are 120–150 d, the annual snow accumulation period is 130–150 d, and the average annual relative humidity is 78%. The main wind direction throughout the year is southwest, with a more southwest wind in spring and summer, and a more northwest wind in autumn and winter.

### 4.2. Sample Collection

Eighty well-grown *P. koraiensis* trees with a diameter of breast height (DBH) of 0.3–100 cm were randomly selected as experimental samples within the broad-leaved and Korean pine climax forest. It was ensured that trees were under similar slope conditions, were in the same slope direction, and were far apart (>10 m). For each sample tree, 3–5 shoots of the same year were randomly selected in the south direction of the upper crown. Soil samples were collected at three points in an equilateral triangle near the sample tree (<1 m) with a sampling shovel. The collection depth was 0–10 cm [51,52]. The soil samples were put into a sealed bag, mixed evenly, and stored in a refrigerated incubator.

### 4.3. Determination of Leaf Traits

In the laboratory, current-year leaves were collected, and the morphological traits including fresh leaf weight, dry leaf weight, leaf volume and needle length were measured. The number of needles (*n*) in the current year was counted, and the fresh weight of leaves was measured by a 10,000-bit electronic balance (accurate to 0.0001 g). The volume of sample leaves (*V*, cm^3^) was determined by the drainage method. Needle length (*l*, cm) was determined using a straightedge with an accuracy of 0.1 cm. Since the cross section of the leaves of *P. koraiensis* is an equilateral triangle and the needles can be seen as triangular prisms, the calculation formula (*A*, cm^2^) for the total epidermis area of *P. koraiensis* leaves [27] is as follows:(1)$$A=\frac{2.28\sqrt{nlV}}{2}$$

The samples were dried in the oven at 65 °C to constant weight, and the dry weight of the leaves was determined (accurate to 0.0001 g). Leaf density (LD) was obtained by the ratio of dry leaf weight to leaf volume, and leaf mass per area (LMA) was obtained by the ratio of leaf dry weight to the total area of leaf epidermis.

The sample leaves were selected and dried in the oven. Leaves were then ground into a fine powder with a grinder, and the ecological stoichiometric characteristics were determined. Leaf carbon content (LC) was determined using a carbon and nitrogen analyzer (multi N/C 3000, Analytik Jena AG, Jena, Germany), and leaf nitrogen content (LN) and leaf phosphorus content (LP) were determined using an automatic discontinuous chemical analyzer (AQ400, SEAL Analytical, Mequon, WI, USA).

A portion of the crushed samples was filtered through mesh (200 mesh) and assayed using the Ecosystem C/N/H/O stable isotope mass spectrometry system (Thermo Fisher Scientific, Bremen, Germany). Leaf δ^13^C (‰) was determined by rapid combustion with the Mat253 + EA-isolink instrument, and δ^18^O (‰) was determined by the high-temperature pyrolysis method with 253 Plus + Flash 2000 HT.

### 4.4. Determination of Abiotic Factors

Under cloudy conditions, hemisphere photography (Nikon Coolpix 4500 digital camera with a 180° fisheye lens, Nikon, Tokyo, Japan) was used to capture hemisphere images [53]. A Gap Light Analyzer (GLA ver. 2.0) [54] was used to calculate the total transmitted incident radiation (including direct and scattered, mol m^−2^ d^−1^) within the zenith angle range of 0–60° for each hemisphere image, and this value was used to characterize the light environment (it is defined as “Light” in all figures and tables).

After soil samples were collected, the fresh weight and dry weight were determined, and the soil water content (SWC) was measured. After drying, the soil was ground into powder with a grinder and filtered through a 100-mesh sieve. Then, 25 g of soil was used to measure the pH using the potentiometric method with a pH meter (pH211, HANNA, Villafranca Padovana, Italy). Soil organic carbon (Soil C) was determined by a carbon and oxygen analyzer (Analytic Jena AG, Jena, Germany), and total nitrogen (N) and total phosphorus (P) were determined by a discontinuous analyzer (AQ400, SEAL Analytical, Mequon, WI, USA).

### 4.5. Data Analysis

Pearson correlation analysis was used to analyze the relationship between δ^13^C, δ^18^O, leaf functional traits, and abiotic factors. Standardized major axis (SMA) analysis was used to determine the correlation between δ^13^C and leaf traits of each DBH group. If the slope was significantly different from 1, an allometric relationship was indicated. The dataset was divided into two groups based on the DBH of the sample tree, and structural equation modeling (SEM) was constructed for each group [28]. Due to the large number of soil physicochemical factors and leaf functional properties, a generalized linear mixed-effects model (GLME) was used to determine the relative importance of each variable on the δ^13^C variation in *P. koraiensis* leaves (Figure A1, Figure A2). Soil physicochemical factors and leaf functional traits with highest relative importance were selected as the variables for optimal SEM model construction, and the direct, indirect, and total effect values of each influencing factor on leaf δ^13^C were calculated.

All data were statistically analyzed by Excel, R 4.1.2, and Origin 2021 and plotted by Sigmaplot 12.0 software. The trend chart was obtained in Excel, the correlation analysis was performed with Origin 2021, and R 4.1.2 was used to generate the standardized major axis analysis (using the “smatr” package), generalized linear mixed-effects model (using the “glmm.hp” package), and the structural equation model (using the “lavaan” package).

## 5. Conclusions

With the growth and development of *P. koraiensis*, plant water use efficiency (WUE) as indicated by δ^13^C increased gradually then leveled off at maturity. The WUE of *P. koraiensis* was primarily affected by photosynthetic carbon sequestration during the entire growth period, and the water use strategy of *P. koraiensis* appeared to be water-saving. The WUE of *P. koraiensis* was regulated by a variety of factors, and the main influencing factors depended on tree size. In the early stages of plant growth (DBH from 0 to 52 cm), DBH contributed the most to δ^13^C, followed by δ^18^O and soil moisture, while light and soil pH showed minimal effect. At plant maturity (DBH from 52 to 100 cm), light environment contributed the most to δ^13^C, followed by soil nitrogen content and soil water content, and leaf nitrogen and DBH contributed little. The present study on the factors influencing WUE during the growth of *P. koraiensis* is helpful in understanding the hydraulic regulation mechanism of plants and the life history strategy of *P. koraiensis*.

## Figures and Tables

**Figure 1 plants-13-01771-f001:**
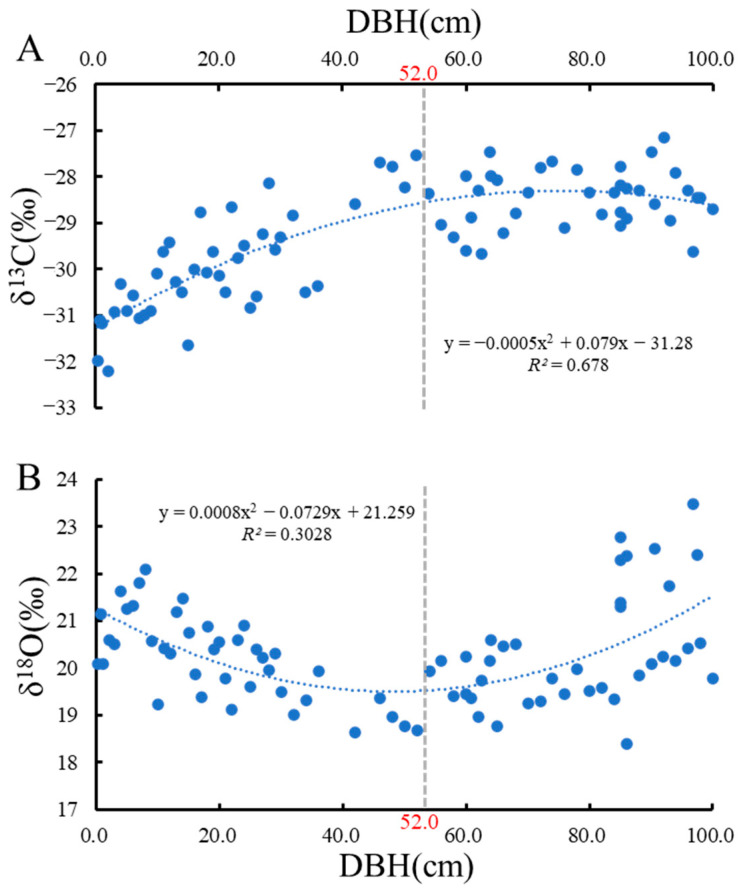
Trend of carbon and oxygen isotopes with DBH. (**A**), Trend of δ^13^C with DBH; (**B**), Trend of δ^18^O with DBH.

**Figure 2 plants-13-01771-f002:**
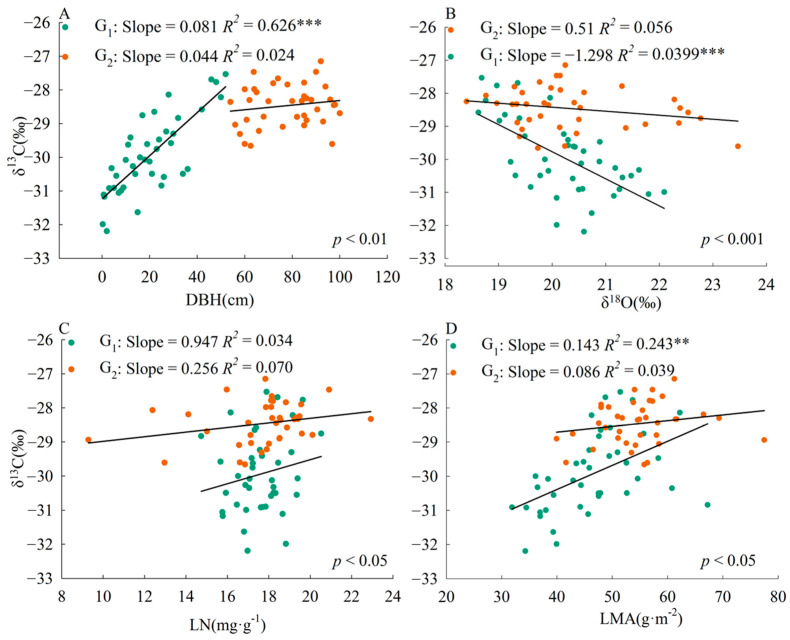
Relationship between leaf δ^13^C and various plant traits in *Pinus koraiensis*. G_1_, DBH is from 0 to 52 cm; G_2_, DBH is from 52 to 100 cm. LN, leaf nitrogen content; LMA, leaf mass per area. The *p*-value indicates the significance of the difference between slopes; *p* < 0.05, significant difference, no common slope; *p* > 0.05, no difference, common slope. **, *p* < 0.01, ***, *p* < 0.001. (**A**), DBH-δ^13^C; (**B**), δ^18^O-δ^13^C; (**C**), LN-δ^13^C; (**D**), LMA-δ^13^C.

**Figure 3 plants-13-01771-f003:**
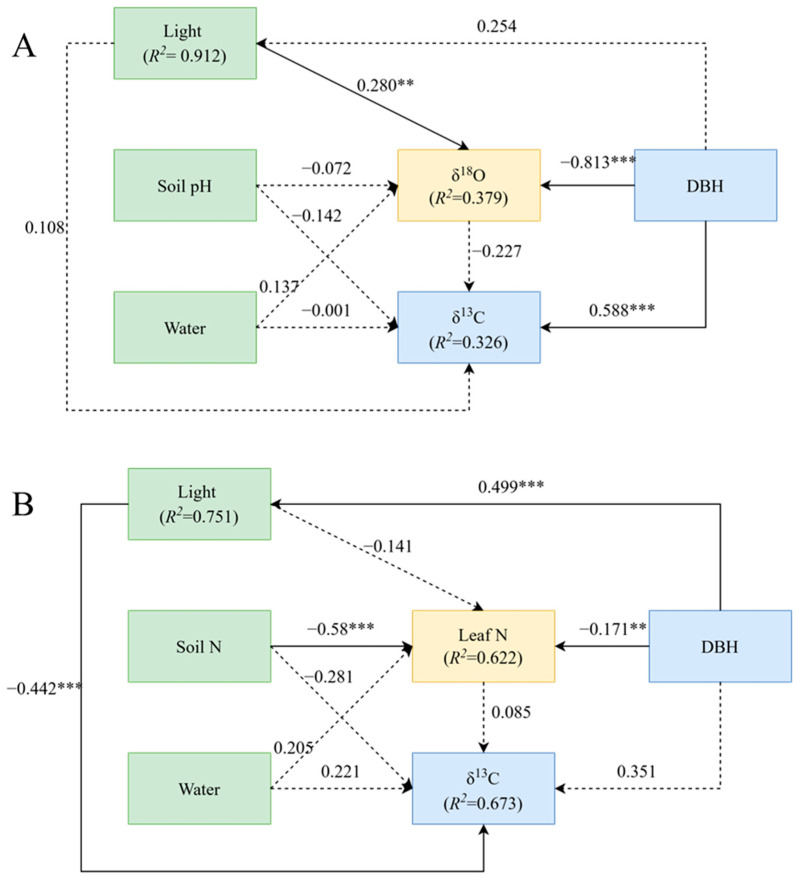
Structural equation model (SEM) between leaf carbon isotopes and their influencing factors. (**A**), The DBH is from 0 to 52 cm with δ^13^C. (**B**), The DBH is from 52 to 100 cm with δ^13^C. ** *p* < 0.01; *** *p* < 0.001.

**Figure 4 plants-13-01771-f004:**
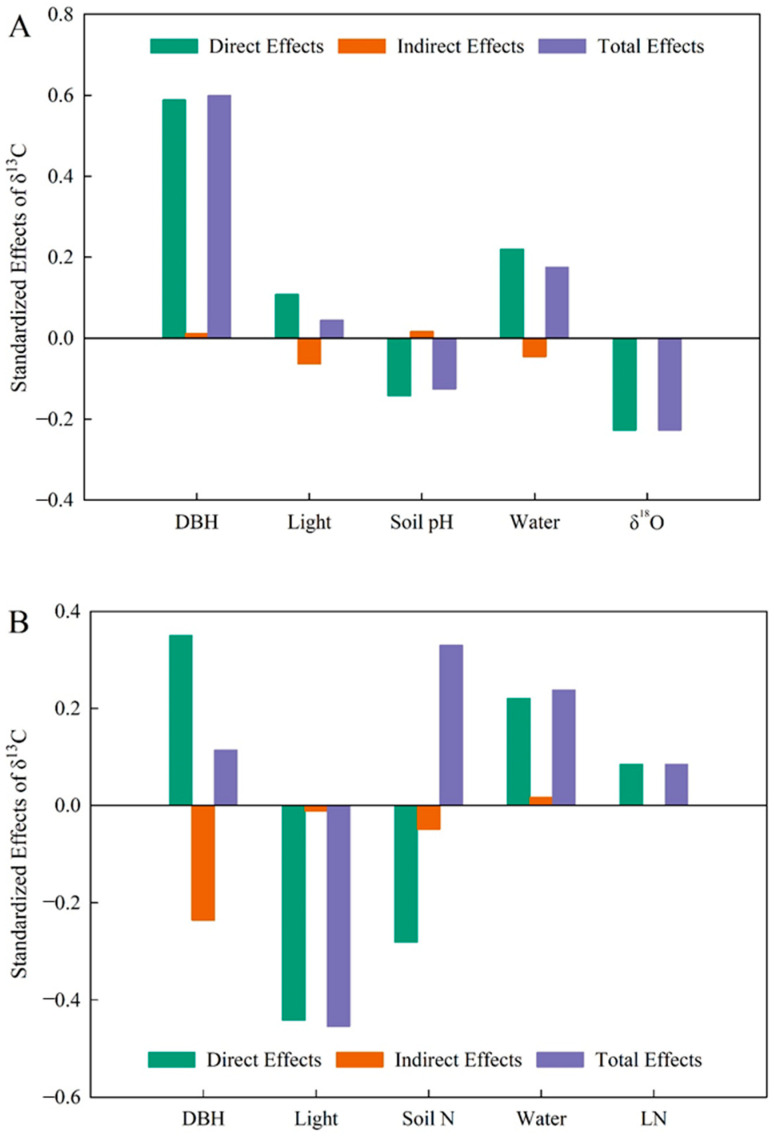
The effect values of various influencing factors on leaf δ^13^C. (**A**), The DBH is from 0 to 52 cm with δ^13^C in *Pinus koraiensis*. (**B**), The DBH is from 52 to 100 cm with δ^13^C in *Pinus koraiensis*. Light, light environment; Water, soil water content.

**Table 1 plants-13-01771-t001:** Correlation analysis between carbon or oxygen isotopes and plant traits (DBH 0–52 cm) LN, leaf nitrogen content; LP, leaf phosphorus content; LC, leaf carbon content; LD, leaf density; LMA, leaf mass per area. * *p* < 0.05, ** *p* < 0.01, *** *p* < 0.001.

	DBH	|δ^13^C|	δ^18^O	LN	LP	LC	LD
|δ^13^C|	−0.76 ***						
δ^18^O	−0.67 ***	0.62 ***					
LN	0.01	−0.19	−0.033				
LP	−0.47 **	0.46 **	0.46 **	−0.18			
LC	−0.26	0.20	−0.01	0.06	0.05		
LD	0.40 *	−0.37 *	−0.24	0.09	−0.32 *	−0.01	
LMA	0.71 ***	−0.59 ***	−0.49 **	0.18	−0.30	−0.01	0.72 ***

**Table 2 plants-13-01771-t002:** Correlation analysis between carbon or oxygen isotopes and abiotic factors (DBH 0–52 cm). Soil pH, soil pH; Water, soil water content; Soil C, soil carbon content; Soil P, soil phosphorus content; Soil N, soil nitrogen content; Light, light environment. * *p* < 0.05, *** *p* < 0.001.

	|δ^13^C|	δ^18^O	Soil pH	Water	Soil C	Soil P	Soil N
δ^18^O	0.62 ***						
Soil pH	0.22	0.01					
Water	0.01	0.01	0.34 *				
Soil C	−0.21	−0.14	0.26	0.73 ***			
Soil P	−0.22	−0.05	0.17	0.63 ***	0.78 ***		
Soil N	−0.22	−0.05	0.17	0.63 ***	0.78 ***	1 ***	
Light	−0.23	0.07	−0.07	−0.10	0.18 ***	0.13	0.13

**Table 3 plants-13-01771-t003:** Correlation analysis between carbon or oxygen isotopes and plant traits (DBH 52–100 cm). LN, leaf nitrogen content; LP, leaf phosphorus content; LC, leaf carbon content; LD, leaf density; LMA, leaf mass per area. * *p* < 0.05, ** *p* < 0.01, *** *p* < 0.001.

	DBH	|δ^13^C|	δ^18^O	LC	LN	LP	LD
|δ^13^C|	0.16						
δ^18^O	0.51 **	−0.18					
LC	−0.07	0.071	−0.35 *				
LN	0.02	0.28	−0.10	0.21			
LP	−0.17	0.18	−0.36 *	0.64 ***	0.45 **		
LD	0.30	0.11	−0.17	0.36 *	−0.30	0.17	
LMA	0.17	0.22	−0.16	0.10	−0.26	−0.01	0.72 ***

**Table 4 plants-13-01771-t004:** Correlation analysis between carbon or oxygen isotopes and abiotic factors (DBH 52–100 cm). Soil pH, soil pH; Water, soil water content; Soil C, soil carbon content; Soil P, soil phosphorus content; Soil N, soil nitrogen content; Light, light environment. * *p* < 0.05, ** *p* < 0.01, *** *p* < 0.001.

	|δ^13^C|	δ^18^O	Soil pH	Water	Soil C	Soil P	Soil N
δ^18^O	0.17						
Soil pH	0.15	0.46 **					
Water	−0.12	−0.58 ***	−0.27				
Soil C	0.14	−0.43 **	−0.23	0.65 ***			
Soil P	−0.22	−0.29	0.05	0.41 *	0.06		
Soil N	0.30	−0.08	0.16	0.22	0.45 **	−0.02	
Light	0.25	0.23	0.08	0.01	−0.09	0.27	−0.01

## Data Availability

The raw data supporting the conclusions of this article will be made available by the authors, without undue reservation.

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
