# Peer review of "Isotope-Based Techniques to Investigate Factors Influencing Water Use Efficiency in Pinus koraiensis Leaves during Plant Growth"

_plants, 2024, doi:10.3390/plants13131771_

Round 1
Reviewer 1 Report
Comments and Suggestions for Authors
The authors of manuscript plants-2998177 presented an interesting study on the correlations between C and O isotope, and tree growth parameters and several environmental factors. δ13C is often used as a indicator for long-term water stress, whereas δ18O can be used to trace water source. However, there was inadequate investigation and data interpretation with regards to these aspects. Some revisions would be required to improve the manuscript.
- Isotope data usually are linked to long-term plant performance. In this study, the authors didn’t monitor long-term radiation level or soil moisture. Instead, transient radiation and soil moisture were collected, seemingly only once. It is unclear when during the day the radiation was measured. Why was the radiation measured above the canopy only? Reference was missing. For soil moisture, one-time measurement in 10cm soil depth would not be able to reflect the soil moisture status in the rhizosphere, and would be affected largely by the last precipitation before the sampling. These types of environmental data are not on the same temporal scale with isotope data.
- As far as WUE is concerned, the study is missing stomatal conductance measurements or any type of leaf gas exchange measurements.
- What is the relation between DBH and root depth in Pinus koraiensis? It would be helpful to add some discussion in this regard. As root depth is relevant to water use and water stress status, it should be a key component in your analysis and data interpretation.
- Fig. 2. DBH52cm seems to be a fairly arbitrary division. Please provide physiological and statistical justification for the 52cm division.
Some minor issues:
Line 24: Please spell out the terms when the abbreviations LP, LMA, LD appear in the text for the first time.
Line 82, 283: Is Pinus koraiensis a broadleaf species?
Author Response
Thank you very much for taking the time to review this manuscript. Please find the detailed responses below and the corresponding revisions in the re-submitted files. Major modifications in the revised manuscript were noted the yellow and we hope this can help you to review our revised manuscript. |
Comments 1: Isotope data usually are linked to long-term plant performance. In this study, the authors didn’t monitor long-term radiation level or soil moisture. Instead, transient radiation and soil moisture were collected, seemingly only once. It is unclear when during the day the radiation was measured. Why was the radiation measured above the canopy only? Reference was missing. For soil moisture, one-time measurement in 10cm soil depth would not be able to reflect the soil moisture status in the rhizosphere, and would be affected largely by the last precipitation before the sampling. These types of environmental data are not on the same temporal scale with isotope data. |
Response 1: Thank you for pointing this out. I have revised the wording regarding light intensity and soil collection, and added corresponding references (L318-L323, L290). The sample collection is in August, when the leaf development is completed, which is the stable period of the leaf canopy of Pinus koraiensis (Wang, B.; Liu, Z.; Qi, Y.; Jin, G. Seasonal dynamics of leaf area index using different methods in the Korean pine plantation. Acta Ecologica Sinica 2014, 34, 1956-1964.). Therefore, the light intensity measured during this time period is stable and representative. To avoid precipitation affecting soil moisture, we choose to collect samples in suitable weather conditions, and the soil samples we took were a mix of three points around the tree. Because most of the fine roots in the broad-leaved Korean pine forest exist within the surface soil layer of 0-10cm (Yang, L.; Li, W. Fine root distribution and turnover in a broad-leaved and Korean pine climax forest of the Changbai Mountain in China. Journal of Beijing Forestry University 2005, 27, 1-5.), we chose to collect them at this soil depth. If soil moisture and light are detected for a long time, it will inevitably be affected by the season, as well as plant leaf growth and withering. During this period, the carbon and oxygen isotopes of plant leaves will also change. Because we were concerned about the state of plant growth in its peak period, we chose a relatively appropriate time for measurement and did not conduct long-term monitoring. However, this opinion is very good, and we will improve it in future research. |
Comments 2: As far as WUE is concerned, the study is missing stomatal conductance measurements or any type of leaf gas exchange measurements. |
Response 2: Thank you for pointing this out. Stomatal conductance and leaf gas exchange data are used to calculate instantaneous water use efficiency. Commonly used transpiration water consumption traits, such as stomatal anatomical structure, may not fully characterize stomatal behavior (Scoffoni, C.; Chatelet, D.S.; Pasquet-kok, J.; Rawls, M.; Donoghue, M.J.; Edwards, E.J.; Sack, L. Hydraulic basis for the evolution of photosynthetic productivity. Nature Plants 2016, 2. )( Pivovaroff, A.L.; Cook, V.M.W.; Santiago, L.S. Stomatal behaviour and stem xylem traits are coordinated for woody plant species under exceptional drought conditions. Plant Cell and Environment 2018, 41, 2617-2626.), while leaves δ18O can provide integrated information on long-term stomatal behavior of leaves, so we use δ18O is used to characterize the comprehensive information of pores. We did not measure stomatal conductance, because the change of stomatal conductance was affected by many factors, but we focused on δ18O, and then said that the index δ18O we measured could represent these. |
Comments 3: What is the relation between DBH and root depth in Pinus koraiensis? It would be helpful to add some discussion in this regard. As root depth is relevant to water use and water stress status, it should be a key component in your analysis and data interpretation. |
Response 3: Thank you for pointing this out. Due to the fact that most of the fine roots in the forest are distributed in the soil surface layer of 0-10cm and Pinus koraiensis of large DBH have a massive root system, it is difficult to measure the root depth. Therefore, we did not collect data in this regard. This opinion is very good, we will consider these factors for measurement and analysis in subsequent experiments. |
Comments 4: Fig. 2. DBH52cm seems to be a fairly arbitrary division. Please provide physiological and statistical justification for the 52cm division. |
Response 4: Thank you for pointing this out. Choosing DBH52cm is partly because the turning point of the “Trend of carbon and oxygen isotopes with DBH”, partly because Pinus koraiensis was classified as a growth and development state before 52cm, and as a mature state after 52cm(Ji, M.; Jin, G.; Liu, Z. Effects of ontogenetic stage and leaf age on leaf functional traits and the relationships between traits in Pinus koraiensis. J For Res (Harbin) 2021, 32, 2459-2471.). This reason has been discussed in the discussion section. |
Comments 5: Line 24: Please spell out the terms when the abbreviations LP, LMA, LD appear in the text for the first time. |
Response 5: Thank you for pointing this out. I have revised this point (L24-L27). |
Comments 6: Line 82, 283: Is Pinus koraiensis a broadleaf species? |
Response 6: Thank you for pointing this out. I have revised this point (L84, L285). |
Reviewer 2 Report
Comments and Suggestions for Authors
The study: “Isotope-based techniques to investigate factors influencing water use efficiency in Pinus koraiensis leaves during plant growth” reports a good set of data related to leaf characteristics and environmental variables for individuals of P. koraiensis of different size. This study is interesting and well described and helps to understand the responses of this species to environmental changes during growth. Other aspects are considered in the following comments, and I have a main concern related to the light measurement.
Abstract. Considering that some acronyms are described in subsequent sections. Please describe LP, LMA, LD and LN here.
L53. WUE.
LL114-115. Describe LP and LMA in this section.
Fig. 3. Describe what SEM means in the caption.
Discussion. Here and elsewhere abbreviate the genus Pinus to P.
L188. According to Fig. 1, plants with DBH under 52 had the isotopic firm of 18O gradually reduced not enhanced.
LL207-209. Place specific leaf weight before the acronym in the first sentence.
LL316-320. The acronym DHP is unnecessary. This measurement is my main concern. From the transmitted radiation for each sampled tree, you get the light above each tree. The units are for day, how come this single measurement can tell you the daily photosynthetic flux? You use a software and get the light intensity for each tree. I think this will depend on the time of the day, on the canopy openness and on the slope where each tree is. I do not buy this variable. Please explain.
Conclusions. Describe WUE here. This is a new section.
Author Response
Thank you very much for taking the time to review this manuscript. Please find the detailed responses below and the corresponding revisions in the re-submitted files. Major modifications in the revised manuscript were noted the yellow and we hope this can help you to review our revised manuscript. |
Comments 1: Abstract. Considering that some acronyms are described in subsequent sections. Please describe LP, LMA, LD and LN here. |
Response 1: Thank you for pointing this out. I have revised this point (L26-L28). |
Comments 2: L53. WUE. |
Response 2: Thank you for pointing this out. I have revised this point (L55). |
Comments 3: LL114-115. Describe LP and LMA in this section. |
Response 3: Thank you for pointing this out. I have revised this point (L116-L117). |
Comments 4: Fig. 3. Describe what SEM means in the caption. |
Response 4: Thank you for pointing this out. I have revised this point (Fig. 3.). |
Comments 5: Discussion. Here and elsewhere abbreviate the genus Pinus to P. |
Response 5: Thank you for pointing this out. I have revised this point (Discussion section). |
Comments 6: L188. According to Fig. 1, plants with DBH under 52 had the isotopic firm of 18O gradually reduced not enhanced. |
Response 6: Thank you for pointing this out. I have revised this point (L190). |
Comments 7: LL207-209. Place specific leaf weight before the acronym in the first sentence. |
Response 7: Thank you for pointing this out. I have revised this point (L209-L211). |
Comments 8: LL316-320. The acronym DHP is unnecessary. This measurement is my main concern. From the transmitted radiation for each sampled tree, you get the light above each tree. The units are for day, how come this single measurement can tell you the daily photosynthetic flux? You use a software and get the light intensity for each tree. I think this will depend on the time of the day, on the canopy openness and on the slope where each tree is. I do not buy this variable. Please explain. |
Response 8: Thank you for pointing this out. I have revised the wording regarding light intensity, and added corresponding references in this section (L318-323). This unit is used because we are measuring photosynthetically active radiation (PAR) rather than photosynthetic photon flux (PPF), and the unit is given in mol·m−2d-1 in the GLAV2 users manual. We drew lessons from this article that Variations in leaf economics spectrum traits for an evergreen coniferous species: tree size dominates over environment factors for this measurement method.
|
Comments 9: Conclusions. Describe WUE here. This is a new section. |
Response 9: Thank you for pointing this out. I have revised this point (L350). |
Round 2
Reviewer 2 Report
Comments and Suggestions for Authors
Abstract. Leaf density: Leaf should not be in capital letter.
Abbreviate the genus name, not the species name. P. koraiensis.
L 321.The units shown for the variable are for photosynthetic photon flux not radiation. I think it does not have anything to do with this study.
Author Response
Thank you very much for taking the time to review this manuscript. Please find the detailed responses below and the corresponding revisions in the re-submitted files. Major modifications in the revised manuscript were noted the yellow and we hope this can help you to review our revised manuscript.
Comments 1: Abstract. Leaf density: Leaf should not be in capital lette. |
Response 1: Thank you for pointing this out. I have revised this point (L25). |
Comments 2: Abbreviate the genus name, not the species name. P. koraiensis. |
Response 2: Thank you for pointing this out. I have revised this point (Discussion section). |
Comments 3: The units shown for the variable are for photosynthetic photon flux not radiation. I think it does not have anything to do with this study. |
Response 3: Thank you for pointing this out. We agree with your point of view that the units shown for the variable are for photosynthetic photon. We use this indicator to characterize light intensity because we consider the impact of plant size and canopy structure on light intensity. And we use this unit because GLA can only export indicators based on this unit. The software manual states that Units are the unit of measure used to represent solar radiation estimates for above and below the forest canopy. In order to calculate the total incident radiation per day, the software sets ideal parameters such as Solar Constant, Kt, Spectral Fraction, Beam Fraction, etc. based on the month of the growing season and location, and uses them to calculate the theoretical value of incident radiation. In article” Celis, J.; Xiao, X.M.; Wagle, P.; Basara, J.; McCarthy, H.; Souza, L. A comparison of moderate and high spatial resolution satellite data for modeling gross primary production and transpiration of native prairie, alfalfa, and winter wheat. Agricultural and Forest Meteorology 2024, 344.”, it is mentioned that “Photosynthetically active radiation (PAR) data were estimated as 0.48 of total incoming shortwave radiation and converted into photosynthetic photon flux density (PPFD) using the approximation 1 W m-2 ≈ 4.57 μmol m-2 s-1”, and in this article, the unit of PAR is mol/m2/day. I hope this answer can be accepted, thanks. |
|
Round 3
Reviewer 2 Report
Comments and Suggestions for Authors
I believed I have not been clear enough in my comments. I apologized and appealed the authors to think a bit about this:
1. in L25 authors mention that “δ13C and δ18O were not related to environmental factors”; however, later they mention that these variables are related to these isotopic signatures (LL27-32).
2. the variable “light intensity” is misleading. What authors measured was the total transmitted incident radiation (LL319-320). Then, the relationships are between the isotopic signatures and canopy openness or leaf area index of the measured trees. Revise.
3. then, this is not an environmental variable.
Author Response
Thank you very much for taking the time to review this manuscript. Please find the detailed responses below and the corresponding revisions in the re-submitted files. Major modifications in the revised manuscript were noted the yellow and we hope this can help you to review our revised manuscript.
|
Point-by-point response to Comments and Suggestions for Authors |
Comments 1: in L25 authors mention that “δ13C and δ18O were not related to environmental factors”; however, later they mention that these variables are related to these isotopic signatures (LL27-32). |
Response 1: Thank you for pointing this out. I have revised this point and deleted the sentence (LL25). |
Comments 2: the variable “light intensity” is misleading. What authors measured was the total transmitted incident radiation (LL319-320). Then, the relationships are between the isotopic signatures and canopy openness or leaf area index of the measured trees. Revise. |
Response 2: Thank you for pointing this out. I am so apologized for not understanding your opinion before. I changed my expression that changing “light intensity” to “light environment” (LL324). I hope this answer can be accepted, thanks. |
Comments 3: then, this is not an environmental variable. |
Response 3: Thank you for pointing this out. Although the measured variable is influenced by the canopy, it also quantitatively reflects the light environment in which the sample branch is located, which we believe is still an environmental variable. I hope this answer can be accepted, thanks. |
Round 4
Reviewer 2 Report
Comments and Suggestions for Authors
I understand you do not want to make more statistics changing this variable, but I do not agree. I am sorry.
Author Response
Thank you very much for taking the time to review this manuscript. Please find the detailed responses below and the corresponding revisions in the re-submitted files. Major modifications in the revised manuscript were noted the yellow and we hope this can help you to review our revised manuscript.
Comments 1: I understand you do not want to make more statistics changing this variable, but I do not agree. I am sorry. |
Response 1: Thank you for pointing this out. We agree with your comments and refer to this factor as abiotic factor. |